# Polycomb group ring finger protein 6 suppresses Myc-induced lymphomagenesis

Nina Tanaskovic[1], Mattia Dalsass[1], Marco Filipuzzi[1], Giorgia Ceccotti[1], Alessandro Verrecchia[1], Paola Nicoli[1], Mirko Doni[1], Daniela Olivero[2], Diego Pasini[1,3], Haruhiko Koseki[4,5], Arianna Sabò[1], Andrea Bisso[1], Bruno Amati[1]

**Max is an obligate dimerization partner for the Myc transcription factors and for several repressors, such as Mnt, Mxd1-4, and Mga, collectively thought to antagonize Myc function in transcription and oncogenesis. Mga, in particular, is part of the variant Polycomb group repressive complex PRC1.6. Here, we show that ablation of the distinct PRC1.6 subunit Pcgf6–but not Mga–accelerates Myc-induced lymphomagenesis in Eμ-*myc* transgenic mice. Unexpectedly, however, Pcgf6 loss shows no significant impact on transcriptional profiles, in neither pre-tumoral B-cells, nor lymphomas. Altogether, these data unravel an unforeseen, Mga- and PRC1.6-independent tumor suppressor activity of Pcgf6.**

## Introduction

The Myc-Max network is constituted by a set of transcription factors that dimerize and bind DNA via a common basic-helix-loop-helix-leucine zipper motif (bHLH-LZ). Max is a key node in this network, acting as an obligate dimerization partner for proteins of the Myc (c-, N- and L-Myc) and Mxd/Mga subfamilies (Mxd1-4, Mnt, and Mga), which activate and repress transcription, respectively, by binding to the same consensus DNA element, the E-box CACGTG and variants thereof (Carroll et al, 2018). Mxd1-4 and Mnt share a short N-terminal domain responsible for recruitment of mSin3/HDAC corepressor complexes. Mga lacks this domain but was independently identified—together with Max—as a component of the variant Polycomb group (PcG) repressive complex PRC1.6, characterized by the presence of two distinct PcG- and E2F-family proteins (respectively, Pcgf6 and E2f6) (Ogawa et al, 2002; Gao et al, 2012; Carroll et al, 2018; Llabata et al, 2020). In mouse embryonic stem cells (mESCs), depletion of Mga led to dissociation of other PRC1.6 subunits (Pcgf6, E2f6 and L3mbtl2) from chromatin (Endoh et al, 2017; Stielow et al, 2018; Scelfo et al, 2019). Along the same line, depletion of Pcgf6 caused dissociation of several subunits (Ring1A/B and Rybp), whereas others (Mga, Max, and L3mbtl2) remained chromatin-bound (Zhao et al, 2017). Altogether, these findings suggest that Mga/Max and Pcgf6 contribute to the hierarchical assembly of the PRC1.6 complex onto chromatin, may thereby counteract transcriptional activation by Myc and E2F at common target genes, and thus also their growth-promoting and oncogenic activities.

A number of observations pointed to a tumor suppressor function of the Mga/Max dimer. First, genome sequencing studies revealed loss-of-function mutations in Mga in a wide variety of tumors (Schaub et al, 2018). Loss of Max was also observed, but appears to be restricted to neuroendocrine tumors, including pheochromocytoma (Comino-Mendez et al, 2011; Burnichon et al, 2012) and small-cell lung cancer (SCLC) (Romero et al, 2014; Llabata et al, 2021). In SCLC, mutations affecting the different network members (Max loss, Mga loss, Myc amplification) occur in a mutually exclusive manner, pointing to a common functional consequence (Romero et al, 2014). Formal evidence for this hypothesis was provided in two SCLC mouse models, in which deletion of Max could either abrogate tumorigenesis if combined with a *MYCL* transgene, or favor it after loss of the Rb1 and Trp53 tumors suppressors (Augert et al, 2020). Hence, in neuroendocrine tumors loss of Mga/Max/PRC1.6 repressor function may be sufficient to bypass the requirement for Myc activity, as recently shown in Max-null human SCLC cell lines (Llabata et al, 2021). In other lineages, the essential role of Max as a Myc partner (Amati et al, 1993) may prevent its loss, but may still co-exist with its antagonist activities in complex with either Mga or Mxd/Mnt proteins. In line with these observations, loss of Mga in a murine Myc-proficient non–small-cell lung cancer model accelerated tumor growth and caused de-repression of PRC1.6, E2F, and Myc/Max target genes (Mathsyaraja et al, 2021).

[1]European Institute of Oncology (IEO) - IRCCS, Milan, Italy    [2]Laboratorio Analisi Veterinarie BiEsseA, A Company of Scil Animal Care Company Srl, Milan, Italy    [3]Department of Health Sciences, University of Milan, Milan, Italy    [4]Laboratory of Developmental Genetics, RIKEN Center for Integrative Medical Sciences, Yokohama, Japan    [5]Cellular and Molecular Medicine, Advanced Research Departments, Graduate School of Medicine, Chiba University, Chiba, Japan

Correspondence: bissoand81@gmail.com; bruno.amati@ieo.it
Nina Tanaskovic's present address is Postbiotica Srl, Milan, Italy
Mattia Dalsass's present address is Department CIBIO, University of Trento, Trento, Italy
Arianna Sabò's present address is QUANTRO Therapeutics GmbH, Vienna, Austria
Andrea Bisso's present address is Gadeta BV, Utrecht, The Netherlands

Recurrent mutations in Mga were also reported in lymphoid malignancies, including Natural Killer/T-cell lymphoma (NKTCL) (Zhang et al, 2018; Kim & Ko, 2022), Chronic Lymphoid Leukemia (Edelmann et al, 2012; De Paoli et al, 2013; Puente et al, 2015) and diffuse large B-cell lymphoma (DLBCL) (Reddy et al, 2017; Lee et al, 2020). Although Myc activation and/or overexpression are widely associated with the progression of these malignancies, it remains to be determined whether Mga and the PRC1.6 complex antagonize Myc activity in this setting.

Here, we addressed whether loss of either Mga or the PRC1.6-restricted subunit Pcgf6 (Gao et al, 2012) potentiate Myc-induced lymphomagenesis in Eμ-*myc* transgenic mice. In previous studies based on the same model, Max was essential for lymphomagenesis (Mathsyaraja et al, 2019); more surprisingly, Mnt also showed tumor-promoting activity in this model, owing most likely to selective suppression of Myc-induced apoptosis (Campbell et al, 2017; Nguyen et al, 2020). Unexpectedly, our data point to a distinct function of Pcgf6 in tumor suppression, independent from either Mga, PRC1.6, or transcriptional control.

# Results and Discussion

### Loss of Pcgf6 accelerates Myc-induced lymphomagenesis

To address the roles of Mga and Pcgf6 in Myc-induced lymphomagenesis, we combined the Eμ-*myc* (Adams et al, 1985) and CD19-*Cre* transgenes (Rickert et al, 1997)—thus expressing both Myc and Cre recombinase from the pro B-cell stage—with either the conditional knockout alleles $Mga^{Inv}$ (hereafter $Mga^{fl}$) (Washkowitz et al, 2015) or $Pcgf6^{fl}$ (Endoh et al, 2017) (Table S1). Whereas targeting *Mga* showed no effect (Fig S1A–C), deletion of *Pcgf6* significantly enhanced Eμ-*myc*–dependent lymphomagenesis, with $Pcgf6^{+/fl}$ and $Pcgf6^{fl/fl}$ animals showing progressive reductions in median disease-free survival, and increased disease penetrance (Fig 1A).

Of note, control Eμ-*myc*;$Mga^{+/+}$ and Eμ-*myc*;$Pcgf6^{+/+}$ animals showed very different kinetics of lymphoma onset, with median disease-free survival times of 203 and 97 d, respectively (Figs 1A and S1A). Such variations are common with the Eμ-*myc* model, whether comparing different studies (e.g., Adams et al [1985], Gorrini et al [2007], and Mathsyaraja et al [2019]). or independent cohorts within the same study as exemplified here, and are most likely attributable to multiple genetic modifiers, especially when the genetic backgrounds intermix as a consequence of the breeding with different alleles. Hence, for each targeted allele, only littermates from the same cohort should be considered as proper "wild-type" controls.

Relative to $Pcgf6^{+/+}$ controls, $Pcgf6^{+/fl}$ and $Pcgf6^{fl/fl}$ tumors (hereafter $Pcgf6^{+/\Delta}$ and $Pcgf6^{\Delta/\Delta}$ or KO) showed proportionate decreases in *Pcgf6* mRNA levels (Fig 1B), and immunoblot analysis confirmed loss of the protein in the latter (Fig 1C). The *Pcgf6* genotype affected neither the differentiation stage of the tumors, with comparable proportions arising from naive mature B-cells (B220+ IgM+) and B-cell precursors (B220+ IgM−) (Fig 1D) (Langdon et al, 1986), nor their pathological classification, all examined cases showing DLBCL/Burkitt's like features (Table S2). Finally, we exploited our RNA-seq profiles (see below) to analyze tumor

clonality through the scoring of reads in the Immunoglobulin heavy chain variable region (Barbosa et al, 2020 Preprint): in contrast with the widespread concept that lymphomas arising in Eμ-*myc* mice are monoclonal, classically based on Southern blotting (Adams et al, 1985) or PCR (Yu & Thomas-Tikhonenko, 2002), we detected multiple clones in most samples (Fig S2 and Table S3), pointing to a more complex oligo- or polyclonal organization of these lymphomas. Most relevant here, our data did not reveal any consistent impact of the *Pcgf6* genotype on clonal complexity, indicating that accelerated tumor onset in the $Pcgf6^{+/f}$ and $Pcgf6^{f/f}$ backgrounds could not simply be ascribed to increased clonality.

Altogether, we conclude that Pcgf6, functions as a dose-dependent, haplo-insufficient tumor suppressor in Myc-induced lymphomagenesis, without altering the gross pathological and cellular features of the resulting tumors. Unlike *Pcgf6*, *Mga* showed no tumor suppressor activity in Eμ-*myc* mice, pointing to a PRC1.6-independent function of Pcgf6 in this model.

### Loss of Pcgf6 affects Myc-induced apoptosis in B-cells

Young Eμ-*myc* mice show a characteristic expansion of pre-tumoral B-cells, counter-balanced by a concomitant increase in apoptosis (Nilsson et al, 2005). Monitoring of bone marrow B220+CD19+ B-cells revealed that their fraction was significantly increased in the $Pcgf6^{f/f}$ background (Fig 2A) correlating with an impairment in Myc-induced apoptosis (Fig 2B). In contrast with the effect on apoptosis, loss of Pcgf6 caused no major alterations in the cell cycle profiles of B220+CD19+ B cells, in either control or Eμ-*myc* transgenic mice (Fig 2C). Of note, whereas the effect of *Pcgf6* loss on Myc-induced lymphomagenesis was already apparent in heterozygous $Pcgf6^{+/fl}$ mutant mice (Fig 1A), the same was not true for B-cell counts and apoptosis (Fig 2A and B): this apparent discrepancy might be due either to a limiting sensitivity of the pre-tumoral measurements, or to the co-existence of additional mechanisms of tumor suppression by Pcgf6. Altogether, our data suggest that the accelerated lymphoma onset in Eμ-*myc*; CD19-*Cre*; $Pcgf6^{fl/fl}$ mice may be explained–at least in part–by increased survival at the pre-tumoral stage, which might favor the expansion of the B220+CD19+ B-cell pool, thus increasing the opportunities for the emergence of tumor clones.

### Loss of Pcgf6 does not affect Myc-dependent transcription

As assayed by RNA-seq profiling, pre-tumoral Eμ-myc B-cells show characteristic changes in gene expression relative to control non-transgenic B-cells (Sabò et al, 2014). This was confirmed in our cohorts, with separate clustering of control and pre-tumoral samples (respectively C and P, Fig 3A); within each cluster, however, the $Pcgf6^{+/+}$ and $Pcgf6^{fl/fl}$ genotypes (WT and KO) remained intermingled. At either the C or P stage, calling for differentially expressed genes (DEGs) between the WT and KO samples yielded virtually no changes (Table S4A–C). Taking WT B-cells as a common control, similar numbers of DEGs were called in WT and KO pre-tumoral samples, with a large overlap between the two genotypes (Fig 3B–D and Table S4D and E). Similarly, RNA-seq profiling of tumor samples (T) yielded high correlation indices among all tumors with no clustering according to Pcgf6 status (Fig S3A), similar

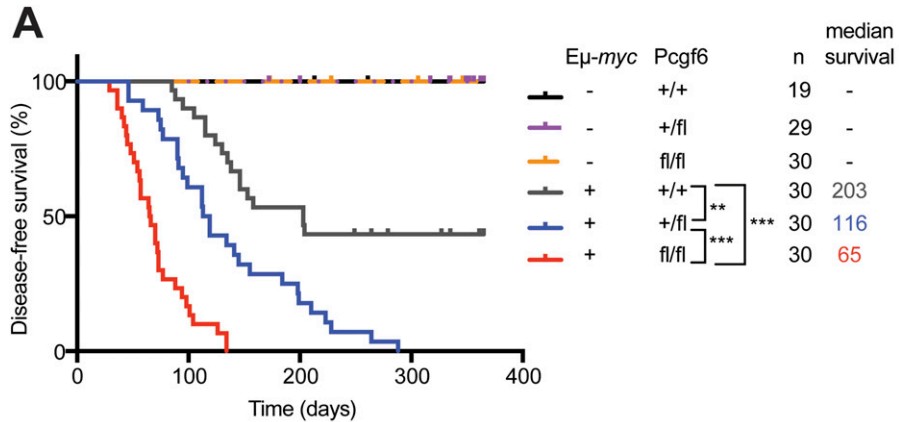

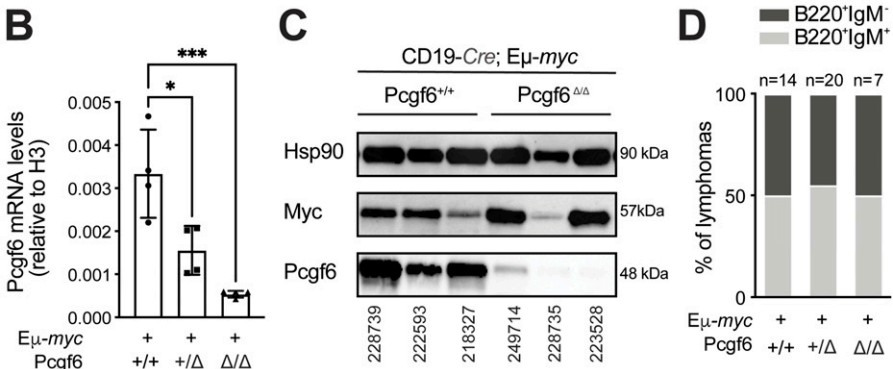

Figure 1. Loss of *Pcgf6* cooperates with Myc overexpression in B-cell lymphoma development. **(A)** Disease-free survival curves for mice of the indicated Eμ-*myc* and *Pcgf6* genotypes (all with the CD19-*Cre* transgene). The number of mice (n) and median survival (in days) are indicated. **(B)** *Pcgf6* mRNA levels were measured by RT-qPCR on mRNA extracted from sorted CD19⁺ lymphoma cells, sampled from infiltrated lymph nodes of CD19-*Cre*; Eμ-*myc* mice, with the indicated *Pcgf6*^fl/fl genotypes. The data show means and s.d.; *$P < 0.05$; **$P < 0.001$; ***$P < 0.0001$. **(C)** Western blot analysis of Pcgf6 and Myc protein expression in infiltrated lymph nodes from either CD19-*Cre*; Eμ-*myc*; *Pcgf6*^+/+ or CD19-*Cre*; Eμ-*myc*; *Pcgf6*^Δ/Δ tumors. Hsp90 was used as loading control. One representative mouse per genotype is shown and mice IDs are indicated at the bottom. Note that a residual Pcgf6 signal in *Pcgf6*^Δ/Δ samples might be due to infiltrating non-deleted cells. **(D)** Immunophenotyping of B220 and IgM reveals similar proportions of B220⁺ IgM⁺ and B220⁺ IgM⁻ tumors among Eμ-*myc* lymphomas of the indicated *Pcgf6* genotypes. The numbers above each bar represent number of mice analyzed for each genotype. Source data are available for this figure.

transcriptional changes in the KO and WT tumors relative to control B-cells (Fig S3B) and very few DEGs (84 up and 81 down) in KO relative to WT tumors (Table S4F and Fig S3C). Most noteworthy here, whereas Pcgf6 was not called as DEG in this comparison, the RNA-seq profiles confirmed the complete absence of Pcgf6 exons 2 and 3 in KO tumors (Fig S3D). In conclusion, Pcgf6 impacted neither on steady-state gene expression profiles, nor on Myc-dependent responses during B-cell lymphomagenesis.

Although few DEGs were called between Pcgf6 KO and WT tumors (Fig S3C), these genes might still be relevant to the more aggressive phenotype of the mutant tumors. Gene Ontology analysis of these DEGs (Fig S3E) pointed out several functional categories, among which we noted several immune-related processes among the down-regulated genes. Whether these reflect true differences in gene expression in Pcgf6 KO versus WT tumor cells or differential infiltration by the host's immune system (e.g., antigen presenting cells, dendritic cells, macrophages, B-, or T-lymphocytes) remains to be determined; nonetheless, these observations suggest that one possible mechanism by which Pcgf6 suppresses lymphomagenesis may be through modulation of anti-tumoral immune responses.

Most noteworthy here, the action of Pcgf6 in Myc-induced lymphoma is opposite to that of Pcgf4 (or Bmi1), a component of the canonical PRC1 complex (Scelfo et al, 2015) that has pro-tumoral activity in Eμ-*myc* mice, mediated by repression of the tumor suppressor locus *Cdkn2a* (or *Ink4/Arf*) (Jacobs et al, 1999). Pcgf6 was

also reported to antagonize the function of another canonical PRC1 subunit, Pcgf2, in anterior-posterior (A-P) specification during embryogenesis (Endoh et al, 2017). By analogy, the tumor suppressor activity or Pcgf6 might have been mediated by suppression of canonical PRC1 activity. However, our RNA-seq data did not support this hypothesis: *Cdkn2a* was expressed at very low levels in control B-cells and was de-repressed in pre-tumoral Eμ-*myc* B-cells, as previously described (Eischen et al, 1999), but loss of Pcgf6 did not reverse this effect (Fig S3F). Together with its limited impact on global expression profiles, these observations suggest that Pcgf6 does not function as an antagonist of canonical PRC1 during lymphomagenesis.

## Mga-dependent recruitment of Pcgf6 to active chromatin

At first sight, the limited impact of Pcgf6 on transcriptional profiles appears at odds with its known function as a component of the PRC1.6 complex. The latter should depend on Mga, which is required both for the integrity of PRC1.6 and for the recruitment of Pcgf6 to chromatin, as shown in mESCs and lung tumor cells (Gao et al, 2012; Endoh et al, 2017; Stielow et al, 2018; Scelfo et al, 2019; Mathsyaraja et al, 2021). To address the status of PRC1.6 in our lymphomas, we derived primary lymphoma cultures from Eμ-*myc* control mice and their *Mga*^−/− and *Pcgf6*^−/− counterparts (Fig S4A). We then used these cells for ChIP-seq profiling of Pcgf6, alongside active histone marks (H3K4me3, H3K4me1, and H3K27ac), as well as the PRC2- and

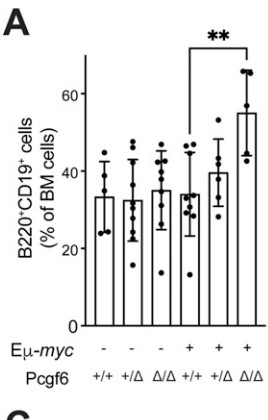

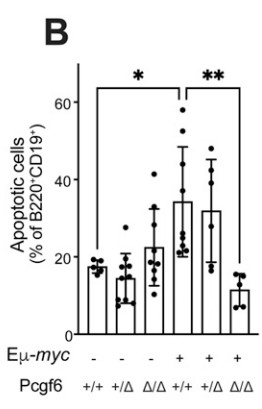

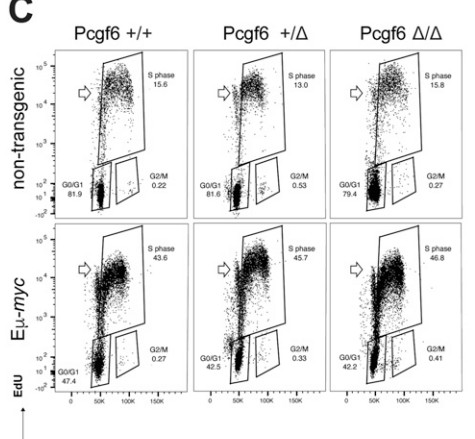

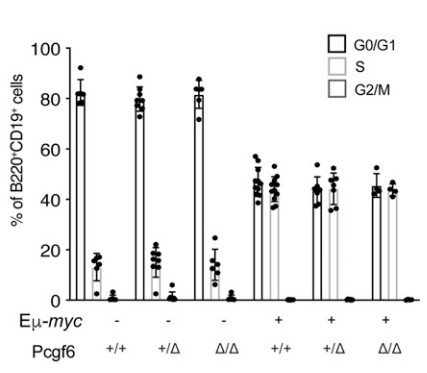

**Figure 2. Pcgf6 loss affects Myc-induced apoptosis, but not proliferation in bone marrow B-cells.**
**(A)** Fraction of B220⁺ CD19⁺ cells in the bone marrow (BM) of the indicated experimental groups. **(B)** Fraction of apoptotic B220⁺ CD19⁺ cells, based on Red-VAD-FMK staining for caspase activity. In both panels, the data show means and s.d.; $*P < 0.05$; $**P < 0.001$. **(C)** Representative EdU/Hoechst flow cytometric profiles of bone marrow–derived B220⁺ CD19⁺ cells in animals of the indicated genotypes. Note that a 2 h EdU pulse in vivo was sufficient for part of the cells to incorporate the nucleotide and complete mitosis, and thus be back in G1 (i.e., with 2N DNA content), whereas scoring as EdU-positive (empty arrowheads): as illustrated here, these cells were neither included in our S-phase gating, nor computed in our G0/G1 counts. Together with the low numbers of cells scored in G2/M, the data imply that B220⁺ CD19⁺ B-cells in vivo undergo mitosis as soon as completing S-phase, with virtually no, or a very short G2 phase. The plot on the right summarizes the data from multiple animals (n ≥ 6).

PRC1-associated repressive marks H3K27me3 and H2AK119Ub (Di Croce & Helin, 2013).

The distribution of ChIP-seq reads among annotated promoters and distal sites in the genome (Fig S4B), confirmed two of key features established in previous studies. First, the Pcgf6 signal observed in the control Eμ-*myc* lymphoma was lost not only in *Pcgf6*⁻/⁻, but also in *Mga*⁻/⁻ cells, in line with the role of Mga in recruiting Pcgf6 to chromatin. Second, Pcfg6 did not co-localize with the PRC-associated marks H3K27me3 and H2AK119Ub but showed preferential binding to active chromatin, as previously observed in mESCs (Stielow et al, 2018; Scelfo et al, 2019). Most relevant here, the propensity to widely associate with active regulatory elements (promoters and enhancers) is a fundamental feature shared by many transcriptional regulators, including Myc/Max and Mga/Max/PRC1.6 complexes (Gao et al, 2012; Sabò et al, 2014; Kress et al, 2016; Endoh et al, 2017; Stielow et al, 2018; Scelfo et al, 2019; Mathsyaraja et al, 2021). As documented for Myc, this initial non-specific engagement of the factor on DNA is insufficient to drive transcription, but is a prerequisite for sequence (i.e., E-box) recognition and establishment of its characteristic gene-regulatory patterns (Sabò & Amati, 2014; Kress et al, 2015; Pellanda et al, 2021). Hence, widespread association with active chromatin—as documented here for Pcgf6—should not be taken to reflect a general role in transcription.

Of note here, one of the apparent changes observed in the *Pcgf6*⁻/⁻ lymphoma was an increase in the H3K27Ac signal on

chromatin, at both proximal and distal sites (Fig S4B). However, owing to the small number of Pcgf6- and Mga-null lymphoma cell lines available in our work, as well as to the limiting availability of compound Eμ-*myc*; *Pcfg6*^fl/fl^ mice (Table S1), which precluded ChIP-seq analysis in pre-tumoral B-cells (Sabò et al, 2014), we could not determine whether this reflected a real effect of Pcg6 on H3K27Ac, or a spurious difference—possibly due to clonal variability among lymphomas. For the same reasons, we were unable to address whether loss of PRC1.6 activity might impact Myc's binding profiles in our model. This scenario appears unlikely, however, given that Pcgf6 loss showed no significant impact on Myc-associated gene expression profiles (Figs 3 and S3).

Altogether, whereas Pcgf6 shows Mga-dependent DNA binding, as expected in the context of the PRC1.6 complex, its deletion does not significantly impact transcriptional programs in either control B-cells, pre-tumoral Eμ-*myc* B-cells, or lymphomas: whether PRC1.6 has a redundant function in transcriptional control or is involved in some other level of chromatin regulation in B-cells remains to be addressed.

## Conclusions and future perspectives

In this work, we unravel a distinct tumor suppressor activity of Pcgf6 in Myc-induced lymphomagenesis, unlinked from Mga and the PRC1.6 complex—and possibly from any direct role in gene regulation. These findings warrant thorough characterization of

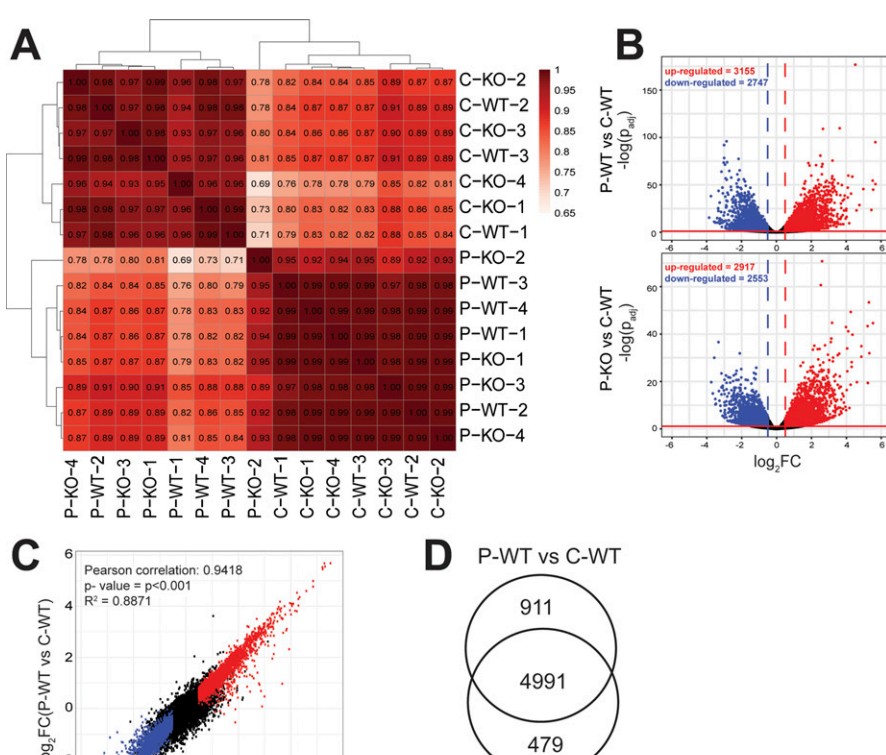

**Figure 3.  Pcgf6 loss does not affect Myc-dependent transcription.**
RNA-seq profiles were generated from control (non-transgenic) and pre-tumoral Eµ-*myc* B-cells (labeled C and P, respectively) with either *Pcgf6*$^{+/+}$ (WT) or *Pcgf6*$^{Δ/Δ}$ (KO) genotypes. All samples are indicated by the stage (C or P) followed by the Pcgf6 genotype (WT or KO) and the sample number. For C-WT, n = 3; C-KO, P-WT, and P-KO, n = 4. **(A)** Pairwise Pearson correlation between all samples, based on their RNA-seq profiles. **(B)** Volcano plots showing the differentially expressed genes (DEGs) called in P-WT (top) or P-KO (bottom) with C-WT as a common control. The horizontal and vertical lines within the plots mark the statistical criteria used for calling DEGs, indicating the thresholds for significance ($P_{adj}$ < 0.05) and fold change (|log$_2$FC > 0.5|). Up- and down-regulated DEGs are shown in red and blue, respectively, and their numbers indicated at the top. All DEGs are listed in Table S4. **(B, C)** Comparison of the DEGs called in P-WT (Y-axis) and P-KO (X-axis), as defined in (B). The DEGs are colored based on the call in the x-axis. $R^2$ = 0.8871 represents the coefficient of determination. **(D)** Venn diagram showing the overlap in DEGs called in P-WT and P-KO.

alternative Pcgf6 activities and of their relevance in human tumors: indeed, besides the PRC1.6 complex, Pcgf6 interacts with the histone H3K4 demethylases JARID1c/d (Lee et al, 2007; Boukhaled et al, 2016) and may have additional partners, yet to be investigated.

Most importantly, our data do not formally rule out a role for Mga/Max and PRC1.6 in antagonizing Myc/Max-dependent transcription in other tumor types, including DLBCL. In particular, the combination of Eµ-*myc* and CD19-*Cre*, targeting *Mga*$^{fl/fl}$, may not reproduce the more mature activated B-cell (ABC) DLBCL subtype in which Mga mutations were reported (Reddy et al, 2017)—although we note that a subset of Eµ-*myc* tumors do show ABC-like expression profiles (Schleich et al, 2020). Moreover, the oncogenic activation of Myc, as modeled by the Eµ-*myc* transgene, might conceivably be sufficient to overcome the repressive function of Mga: in ABC-type DLBCL, in which *MYC* translocation is relatively infrequent (Reddy et al, 2017; Bisso et al, 2019), Myc activity may well be antagonized by Mga, underlying the selective pressure to inactivate it. Resolving this question would imply the joint scoring of *MYC* translocations and Mga mutations in a sizeable number of DLBCL samples, well above those reported so far (125 cases, of which 42 with MYC rearrangements and 7 with MGA mutations) (Reddy et al, 2017). Finally, any of the five Mxd/Mnt proteins that form alternative dimers with Max may also contribute repressive activity on common Mga- and Myc-target genes, and the balance between all these factors may differ between cell/tumor subtypes, experimental models and/or clinical cases.

Altogether, the contribution of the Mga/Max-PRC1.6 complex to DLBCL pathogenesis remains to be addressed. This notwithstanding, our data in the Eµ-*myc* model establish that in conditions in which Mga shows no obvious impact, Pcgf6 deletion clearly accelerates Myc-induced lymphomagenesis.

# Materials and Methods

### Mouse strains and genotyping

Mice bearing the conditional allele *Mga*$^{fl}$ (originally called *Mga*$^{Inv}$) (Washkowitz et al, 2015) were bred with either CD19-*Cre* (Rickert et al, 1997) (a gift of Klaus Rajewsky) or Eµ-*myc* transgenic animals (Adams et al, 1985), and the resulting compound mice bred to obtain the *Mga*-targeted cohort. The same strategy was pursued with the *Pcgf6*$^{fl}$ allele (Endoh et al, 2017). The final crosses used to obtain our experimental cohorts are reported in Table S1. Of note, the *Pcgf6*$^{fl}$ cohort was inbred C57BL/6J, whereas the *Mga*$^{fl/fl}$ cohort was of mixed genetic background (Washkowitz et al, 2015). In all experiments, gender- and age-matched mice (both females and males) were used without randomization or blinding. Genomic DNA extraction and genotyping were performed as previously described (Bisso et al, 2020), with the PCR primers listed in Table S5.

Eμ-*myc* transgenic mice were monitored two to three times a week for tumor development by visual inspection and peripheral lymph node palpation, and were euthanized as soon as they showed signs of lymphoma (i.e., enlarged lymph nodes) (Adams et al, 1985). For pre-tumoral analysis, mice were collected at 4–6 wk of age: spleen and bone marrow were dissected and processed for molecular analysis as previously described (Campaner et al, 2010).

Experiments involving animals were carried out in accordance with the Italian Laws (D.lgs. 26/2014), which enforces Dir. 2010/63/EU (Directive 2010/63/EU of the European Parliament and of the Council of 22 September 2010 on the protection of animals used for scientific purposes) and authorized by the Italian Minister of Health with projects 391/2018-PR.

### Isolation and culturing of primary murine lymphoma cell lines

Mice were inspected personally for tumor development. Infiltrated lymph nodes, spleen and bone marrow were collected and smashed in PBS. Cell suspensions were passed three times through a Falcon 70 μm Cell Strainer (#352350; Corning), centrifuged (80*g* for 5 min) and resuspended in 10 ml of Erythrocyte Lysis buffer (150 mM $NH_4Cl$, 10 mM $KHCO_3$, and 0.1 mM EDTA). After another centrifugation step, cells were resuspended in 10 ml of MACS buffer (PBS, 2 mM EDTA, and 0.5% BSA), and part of the cells used for in vitro culture. Primary cells were grown in suspension in B-cell medium composed of a 1:1 ratio of DMEM (ECM0103L; Euroclone) and IMDM (I3390; Sigma-Aldrich), supplemented with 10% fetal calf serum (Globefarm Ltd.), 2 mM L-glutamine (Invitrogen Life Technologies), 1% non-essential amino acids (NEAAs), 1% penicillin/streptomycin and 25 μM $\beta$-mercaptoethanol. A lymphoma cell line was considered as stabilized when the splitting ratio reached 1:10 every 2 d, which usually occurred upon 2 wk of in vitro culture.

### Analysis of apoptosis, proliferation, and surface markers

Apoptosis in bone marrow–derived B-cells was measured with the CaspGLOW Red Active Caspase Staining Kit (#K190; BioVision) following the manufacturer's guidelines. Proliferation was quantified by EdU staining: EdU (#A10044; Invitrogen) was dissolved in sterile PBS to a concentration of 5 mg/ml; for in vivo proliferation studies, 1 mg EdU in a volume of 200 μl was injected intraperitoneally 2 h before analysis, followed by staining with the 647 EdU Click Proliferation kit (#565456; BD Pharmingen) according to manufacturer's guidelines. Samples were stained with Hoechst DNA content dye, acquired on a FACSCelesta cytofluorimeter, and analyzed using FlowJo Version 10.4.0 software.

For staining of surface markers, cells were incubated in MACS buffer with fluorochrome-conjugated antibodies (used at the dilutions indicated in Table S5) for at least 1 h at 4°C in the dark, and analyzed by flow cytometry, as above.

### Immunoblotting

Protein extraction and immunoblotting were performed as previously described (Bisso et al, 2020) with the indicated primary antibodies (Table S5).

### Hematoxylin and Eosin staining

For hematoxylin and eosin staining and pathological analysis tissues were collected and processed as follows. Freshly isolated lymphoma samples were washed in PBS, fixed in 4% (vol/vol) paraformaldehyde at 4°C degrees for at least 16–24 h, washed in PBS, and stored in 70% ethanol at 4°C for a maximum of 1 wk before inclusion. For the latter, each tissue was dehydrated with increasing concentrations of ethanol, embedded in paraffin blocks and stored at RT. For hematoxylin and eosin staining each block was cut into 3/5-mm thick sections and mounted on glass slides. Slides were counterstained with Harris Hematoxylin (#HHS80; Sigma-Aldrich) and Eosin Y solution (#HT110216; Sigma-Aldrich), dehydrated through alcoholic scale, and mounted with Eukitt (#09-00250; Bio-Optica). All images were acquired with the Aperio Digital Pathology Slide Scanner ScanScopeXT (Leica) before pathological evaluation.

### RNA sequencing

RNA extraction, processing, and sequencing, as well as the filtering of RNA-seq reads and bioinformatic and statistical analyses, were performed as previously described (Tesi et al, 2019; Bisso et al, 2020; Pellanda et al, 2021). The analysis of tumor clonality from RNA-seq reads was performed as previously described (Barbosa et al, 2020 *Preprint*).

### ChIP sequencing

The fixation of in vitro stabilized lymphoma cell lines and their processing for chromatin immunoprecipitation (ChIP) was performed as previously described (Sabò et al, 2014). 5 μg of each of the antibodies listed in Table S5 were used to immunoprecipitate either 500 μg (for the mapping of Myc, Max and Pcgf6) or 250 μg of fixed chromatin (for the histone marks H3K4me3, H3K4me1, H3K27ac, H3K27me3, and H2Ak119Ub). Whereas Myc and Max precipitates were processed exactly as in Sabò et al (2014), Pcgf6 and histone mark precipitates were processed as in Scelfo et al (2019). 1.5–2 ng of DNA was then used to generate the chromatin immunoprecipitation sequencing (ChIP-Seq) libraries according to the Illumina protocol, and sequenced with the Illumina NovaSeq 6000.

ChIP-seq reads were analyzed as previously published (Sabò et al, 2014; Pellanda et al, 2021). Peaks were mapped and annotated according to the genomic position of their midpoint, as (i) promoter: between −2 and +1 Kb from the annotated refgene start coordinate or transcriptional start site (TSS); (ii) gene body: between >1 Kb from the TSS to the 3′ end of an annotated refgene; (iii) intergenic: all peaks positioned outside of the aforementioned intervals. Qualitative and quantitative heatmaps of ChIP-seq enrichment were generated using R with Bioconductor and compEpiTools packages, tools for computational epigenomics (Gentleman et al, 2004; Kishore et al, 2015).

### Oligonucleotide primers

Primers for mRNA analysis were designed with Primer-BLAST (https://www.ncbi.nlm.nih.gov/tools/primer-blast/) (Ye et al,

2012). The complete list of primers used in this study is shown in Table S5.

## Statistical analysis

All experiments were performed at least in biological triplicates. Sample size was not predetermined but is reported in the respective Figure legends. $P$-values were calculated with one-way ANOVA using Tukey correction, except in Fig 1A for Kaplan–Meier survival curves where log p-rank test was used.

# Data Availability

The RNA-seq data produced in this work have been deposited in NCBI's Gene Expression Omnibus (https://www.ncbi.nlm.nih.gov/geo/) and are accessible through the GEO Series accession number GSE190000.

# Supplementary Information

# Acknowledgements

We thank Stefano Campaner, Francesco Nicassio, Diego Pasini, and members of the Amati lab for discussions, insight, suggestions, and reagents; A Gobbi, M Capillo, and all members of the animal facility for their help with the management of mouse colonies; S Bianchi, L Rotta, T Capra, and L Massimiliano for assistance with Illumina sequencing; S Ronzoni for assistance with flow cytometry; MG Jodice, F Montani, G Bertalot, and S Pece for the help with processing tissue samples; VE Papaioannou for providing $Mga^{fl/fl}$ mice; and K Rajewsky for CD19-$Cre$ mice. This work was supported by grants from the Italian Health Ministry (RF-2011-02346976), from the Italian Association for Cancer Research (AIRC, IG2015-16768, and IG2018-21594) to B Amati, and from the Ministry of Education, Culture, Sports, Science and Technology of Japan (Grants-in-Aid for Scientific Research, 23249015) to H Kozeki.

## Author Contributions

N Tanaskovic: formal analysis, validation, investigation, visualization, and writing—original draft, review, and editing.
M Dalsass: data curation, formal analysis, and visualization.
M Filipuzzi: data curation, formal analysis, visualization, and writing—review and editing.
G Ceccotti: investigation.
A Verrecchia: provided technical support.
P Nicoli: provided technical support.
M Doni: provided technical support.
D Olivero: formal analysis and pathological analysis.
D Pasini: provided Pcgf6-mutant mice.
H Koseki: provided Pcgf6-mutant mice.
A Sabo: conceptualization, formal analysis, and supervision.
A Bisso: conceptualization, formal analysis, supervision, investigation, and writing—original draft, review, and editing.
B Amati: conceptualization, supervision, funding acquisition, project administration, and writing—original draft, review, and editing.

## Conflict of Interest Statement

The authors declare that they have no conflict of interest.

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
