## [Reviewer comments · Life Science Alliance]

Life Science Alliance

Polycomb Group Ring Finger Protein 6 suppresses Myc-induced lymphomagenesis

Nina Tanaskovic, Mattia Dalsass, Marco Filipuzzi, Giorgia Ceccotti, Alessandro Verrecchia, Paola Nicoli, Mirko Doni, Daniela Olivero, Diego Pasini, Haruhiko Koseki, Arianna Sabo, Andrea Bisso, and Bruno Amati

DOI: <https://doi.org/10.26508/lsa.202101344>

Corresponding author(s): Bruno Amati, European Institute of Oncology (IEO) and Andrea Bisso, European Institute of Oncology (IEO)

Review Timeline:

Submission Date:	2021-12-17
Editorial Decision:	2022-01-21
Revision Received:	2022-03-10
Editorial Decision:	2022-03-29
Revision Received:	2022-04-04
Accepted:	2022-04-04

Transaction Report:

January 21, 2022

Re: Life Science Alliance manuscript #LSA-2021-01344-T

Bruno Amati
European Institute of Oncology (IEO)
Dept. of Molecular Oncology and Center for Genomic Science
Via Adamello, 16
Milan, Milan 20139
ITALY

Dear Dr. Amati,

Thank you for submitting your manuscript entitled "Polycomb Group Ring Finger Protein 6 suppresses Myc-induced lymphomagenesis" to Life Science Alliance. The manuscript was assessed by expert reviewers, whose comments are appended to this letter. We invite you to submit a revised manuscript addressing the Reviewer comments.

Thank you for this interesting contribution to Life Science Alliance. We are looking forward to receiving your revised manuscript.

Sincerely,

B. MANUSCRIPT ORGANIZATION AND FORMATTING:

Reviewer #1 (Comments to the Authors (Required)):

In this report the authors seek to extend recent studies on the role of the MGA protein in tumor suppression. MGA is a member of the MYC bHLHZ network which dimerizes with MAX and also acts as a scaffold for the assembly of the non-canonical PRC1.6 complex. The latter is comprised of multiple subunits including PCGF6, E2F6, L3MBTL2 and others. MGA inactivation has been linked to small cell lung cancer and to a wide range of other tumor types, notably diffuse large B cell lymphoma (DLBCL). Here the authors begin by using the Eu-myc murine B cell lymphoma model to determine how MGA loss affects lymphoma progression - they do not find evidence that MGA loss influences tumor-free survival of Eu-myc mice (however see point 1 below).

They then focus on one of the defining subunits of the PRC1.6 complex - they conditionally inactivate PCGF6 and demonstrate that its loss significantly accelerates Eu-myc lymphomagenesis, most likely by increasing survival of premalignant B cells. Surprisingly both expression profiling and genomic binding/epigenetic analysis failed to yield clear differences between Eu-myc and Eu-myc; Pcgf6 inactivated B cells or tumors. This is puzzling in that it doesn't align with what might be expected from a model of neoplasia driven by alterations in transcriptional regulatory proteins. The authors might consider comparative proteomic analysis of the tumors.

While the lack of mechanistic data leaves unclear the precise pathways involved, I feel that once the comments below are addressed, the paper will warrant publication since it points to a novel potential tumor suppressor activity of Pcgf6 that may be independent of transcriptional alterations. While the results, as they stand, are phenomenological, they are nonetheless intriguing and may inform and stimulate further studies on the complex role of MGA/PRC1.6 in lymphomagenesis.

Specific comments:

1. Fig 1 and Fig S1: The timing of lymphoma onset in the Eu-myc; Mga+/+ and the Eu-myc; Pcgf6+/+ are significantly different (median survival 97 days vs 203 days, respectively). The authors explain the difference in the figure legend (Fig S1) as most likely being due to different genetic background of the two cohorts. Yet, the problem remains in that the much more accelerated tumorigenesis in the Mga cohort may mask any accelerating effects of Mga loss. This would appear to seriously complicate the interpretation of the data relating to the conclusion regarding Mga inactivation. I think that the statement on page 8 that "Mga shows no impact" in the lymphoma model needs to be qualified.
2. While the authors mention that recurrent Mga mutations in DLBCL it's not clear whether Mga is mutated in the less mature pre-B cell lymphomas that are predominantly modeled by the Eu-myc mice. The authors should clarify this in the discussion.

Reviewer #2 (Comments to the Authors (Required)):

In this manuscript, Tanaskovic et al use genetically engineered mouse models to investigate the contributions of Mga and Pcgf6 to Myc-dependent lymphomagenesis in the Emu-Myc model. These two factors are part of the variant Polycomb group repressive complex PRC1.6. The PRC1.6 complex contains Max, an obligate heterodimerization partner of Myc, and binds to the same E-box motifs as Myc-Max complexes to repress transcription. Mga and Max have been found to play tumor suppressor roles in some models, although this seems to be contextual and probably modified by the presence of other Myc-Max network factors, for example.

The main claim of this manuscript is that in the Emu-Myc model, Pcgf6 (but not Mga) works as a tumor suppressor. A few mechanistic models were explored through a quite detailed and thorough set of experiments, yielding mostly negative results (no major transcriptional changes, no major effects on cell cycle, no major epigenetic changes). In lack of a proven mechanism, the conclusion is that Pcgf6's tumor suppressor effects are disconnected from its known role as part of PRC1.6 complexes. This, by itself, would be an interesting finding.

The differences in tumor incidence and latency across the different models are central to support the authors' claim that Pcgf6 works as a tumor suppressor. Also, to justify the search for a molecular mechanism. However, I find some inconsistencies in these data, which may reflect technical issues (although I am not sure). Confirming the in vivo phenotypes would thus be very important.

Finally, the already thorough genomic and molecular analyses could benefit from some further data analysis and additional experiments to understand how disruption of Pcgf6 alters (or not) the PRC1.6 complex in this model.

These are my detailed comments:

1- The survival curves for Emu-Myc mice (only) in Fig 1 and Fig S1 look quite different. For example, the median survival is 97 days in Fig S1A (in the experiment comparing survival in the Mga model), while it is 203 days in Fig 1 (Pcgf6 model). Instead, the median survival for the combinations with Mga versus Pcgf6 loss are quite similar, and perhaps statistically not significant. The median survival for the Emu-Myc model that can be inferred from other publications is also variable. Resolving the discrepancy in the Emu-Myc reference model is critical to claim that Pcgf6 is working as a tumor suppressor.

2- Minor: The analysis on potential differences in clonality as a measure of increased tumor initiation (Fig S2) is interesting but raises some questions. For example, the possibility that the percentage of purity in each tumor may be affecting the result. Was this taken into account? One could argue that these tumors are still oligoclonal.

3- Can the authors provide more information about the 78+65 differentially expressed genes in the tumor RNA-Seq analysis? This gene list doesn't seem to have been included in the Supplementary tables, and one wonders if using for example pathway analysis tools would yield any relevant findings.

Also, restricting the differential analysis to those genes that are direct targets of Pcgf6 (based on the ChIP-Seq data generated) could yield different results.

4- It would be important to include in the discussion the fact that in DLBCL, Mga seems to score only as a potential tumor suppressor in ABC-like DLBCL, but not GCB-like. Emu-Myc models do not resemble DLBCL in major genetic or phenotypic traits, but if any, the resemblance to ABC-like DLBCL may be smaller than GCB.

5- The effects of Pcgf6 loss on Myc-dependent apoptosis in Emu-Myc B cells (Fig. 2) do not seem associated with haploinsufficiency, this is, there is no loss of apoptosis in Pcgf6 del/+ B cells despite these animals have increased tumor incidence and shortened latency. That would mean that this is perhaps not a major mechanism by which Pcgf6 loss contributes to tumorigenesis (which is opposite to the author's argument).

6- In the same Fig 2, the authors show data on cell cycle analysis. There do not seem to be major differences, but one wonders if the fraction of cells in cycle (Ki67+) is affected by Pcgf6 loss (since many B cells in the bone marrow are resting).

It would also be important to show representative cell cycle plots to help understand why the low percentage of cells in G2/M.

7- An important part of this study is the comparison between the consequences of Mga and Pcgf6 loss. These two factors are components of the PRC1.6 complex. Loss of Mga disrupts the complex (previous studies and this one) and prevents Pcgf6 binding to chromatin. It is unclear, however, what happens to Mga when Pcgf6 is lost. Zhao et al (J Biol Chem 2017, 292) reported that in ES cells, loss of Pcgf6 results in a partial disruption of the PRC1.6 complex, although Mga, Max and L3mbtl2 remain bound to chromatin in a smaller complex. There were also some specific changes in gene expression but no effects on H2AK119Ub levels. Based on these observations and the current data, it would be important to investigate the composition of the PRC1.6 complex in the Emu-Myc system upon Pcgf6 is lost (e.g., co-IP). Complementary to the above, it would be important to determine the distribution of Mga on chromatin (at least via immunoblot of chromatin-enriched fractions, if ChIP-Seq is not feasible). This would be a helpful molecular insight to understand the different outcomes of Mga and Pcgf6 loss in vivo.

8- ChIP-Seq data (Fig S4): the data shows that loss of Pcgf6 results in a (significant) increase in H3K27Ac reads. The authors do not discuss this in the manuscript, but this effect seems quite prominent, extends to both promoters and distal sites, and may have some conceptual implications. Any insights?

Life Science Alliance, manuscript LSA-2021-01344-T

Tanaskovic et al.

Polycomb Group Ring Finger Protein 6 suppresses Myc-induced lymphomagenesis

Authors' point by point rebuttal

Reviewer #1 (Comments to the Authors (Required)):

In this report the authors seek to extend recent studies on the role of the MGA protein in tumor suppression. MGA is a member of the MYC bHLHZ network which dimerizes with MAX and also acts as a scaffold for the assembly of the non-canonical PRC1.6 complex. The latter is comprised of multiple subunits including PCGF6, E2F6, L3MBTL2 and others. MGA inactivation has been linked to small cell lung cancer and to a wide range of other tumor types, notably diffuse large B cell lymphoma (DLBCL). Here the authors begin by using the Eu-myc murine B cell lymphoma model to determine how MGA loss affects lymphoma progression - they do not find evidence that MGA loss influences tumor-free survival of Eu-myc mice (however see point 1 below).

They then focus on one of the defining subunits of the PRC1.6 complex - they conditionally inactivate PCGF6 and demonstrate that its loss significantly accelerates Eu-myc lymphomagenesis, most likely by increasing survival of premalignant B cells. Surprisingly both expression profiling and genomic binding/epigenetic analysis failed to yield clear differences between Eu-myc and Eu-myc; Pcgf6 inactivated B cells or tumors. This is puzzling in that it doesn't align with what might be expected from a model of neoplasia driven by alterations in transcriptional regulatory proteins. The authors might consider comparative proteomic analysis of the tumors.

While we agree with the Reviewer that comparative proteomics or other levels of analysis would be relevant to address the mechanisms of tumor suppression by Pcgf6, we believe that these studies are beyond the scope of the present paper (please see the next point).

While the lack of mechanistic data leaves unclear the precise pathways involved, I feel that once the comments below are addressed, the paper will warrant publication since it points to a novel potential tumor suppressor activity of Pcgf6 that may be independent of transcriptional alterations. While the results, as they stand, are phenomenological, they are nonetheless intriguing and may inform and stimulate further studies on the complex role of MGA/PRC1.6 in lymphomagenesis.

We thank the Reviewer for making this point, which precisely reflects our mindset in this submission: with this work, we feel that we have closed the full cycle of experiments that were needed to address our initial hypotheses, namely (i.) the postulated tumor-suppressive activity of the PRC1.6 complex (in particular MGA and PRC1.6), and (ii.) dissection of the underlying gene expression programs.

The results are unexpected, and point to a different and more complex scenario, given the PRC1.6- and MGA-independent tumor suppressor activity of PCGF6 unraveled by our data. While a number of other experiments should be undertaken to characterize this further, we feel that these belong to future endeavors. The main point here is to report our findings to the scientific community, in order to foster and motivate follow-up studies.

We wish to highlight here that our data do not rule out a possible role of MGA and the PRC1.6 complex in antagonizing MYC activity and/or suppressing lymphomagenesis in specific settings, but provide a clear example that these activities may not always be rate-limiting, and/or may be superseded in specific contexts (e. g. direct activation of MYC itself, as in our work). These aspects are duly accounted for in our text: "Most importantly, our data do not formally rule out a role for Mga/Max and PRC1.6 in antagonizing Myc/Max-dependent transcription in other tumor types, including DLBCL. In particular, etc..."

Specific comments:

1. Fig 1 and Fig S1: The timing of lymphoma onset in the Eu-myc; Mga+/+ and the Eu-myc; Pcgf6+/+ are significantly different (median survival 97 days vs 203 days, respectively). The authors explain the difference in the figure legend (Fig S1) as most likely being due to different genetic background of the two cohorts. Yet, the problem remains in that the much more accelerated tumorigenesis in the Mga cohort may mask any accelerating effects of Mga loss. This would appear to seriously complicate the interpretation of the data relating to the conclusion regarding Mga inactivation. I think that the statement on page 8 that "Mga shows no impact" in the lymphoma model needs to be qualified.

This is an important point, which was pointed out by both Reviewers, and was not explicit enough in our original manuscript. We realize that the note provided in the legend to Fig. S1 was insufficient to clarify this issue. This is now addressed directly in our main text:

- "Of note, control $E\mu\text{-myc};Mga^{+/+}$ and $E\mu\text{-myc};Pcgf6^{+/+}$ animals showed very different kinetics of lymphoma onset, with median disease-free survival times of 203 and 97 days, respectively (Fig. 1A, Fig. S1A). Such variations are common with the $E\mu\text{-myc}$ model, whether comparing different studies (e. g. Adams et al. 1985; Gorrini et al. 2007; Mathysaraja et al. 2019) or independent cohorts within the same study as exemplified here, and are most likely attributable to multiple genetic modifiers, especially when the genetic backgrounds intermix as a consequence of the breeding with different alleles. Hence, for each targeted allele, only littermates from the same cohort should be considered as proper "wild-type" controls."
- The passage quoted by the Reviewer was re-phrased as follows: "... Altogether, the contribution of the Mga/Max-PRC1.6 complex to DLBCL pathogenesis remains to be addressed. This notwithstanding, our data in the $E\mu\text{-myc}$ model establish that in conditions in which Mga shows no obvious impact, Pcgf6 deletion clearly accelerates Myc-induced lymphomagenesis"

2. While the authors mention that recurrent Mga mutations in DLBCL it's not clear whether Mga is mutated in the less mature pre-B cell lymphomas that are predominantly modeled by the $E\mu\text{-myc}$ mice. The authors should clarify this in the discussion.

This is indeed an interesting point that was also brought up by Reviewer #2, and was missing in our manuscript. We now have completed our text to address this: for full description, please refer to our answer to Reviewer #2 (point 4).

Reviewer #2 (Comments to the Authors (Required)):

In this manuscript, Tanaskovic et al use genetically engineered mouse models to investigate the contributions of Mga and Pcgf6 to Myc-dependent lymphomagenesis in the $E\mu\text{-Myc}$ model. These two factors are part of the variant Polycomb group repressive complex PRC1.6. The PRC1.6 complex contains Max, an obligate heterodimerization partner of Myc, and binds to the same E-box motifs as Myc-Max complexes to repress transcription. Mga and Max have been found to play tumor suppressor roles in some models, although this seems to be contextual and probably modified by the presence of other Myc-Max network factors, for example.

The main claim of this manuscript is that in the $E\mu\text{-Myc}$ model, Pcgf6 (but not Mga) works as a tumor suppressor. A few mechanistic models were explored through a quite detailed and thorough set of experiments, yielding mostly negative results (no major transcriptional changes, no major effects on cell cycle, no major epigenetic changes). In lack of a proven mechanism, the conclusion is that Pcgf6's tumor suppressor effects are disconnected from its known role as part of PRC1.6 complexes. This, by itself, would be an interesting finding.

The differences in tumor incidence and latency across the different models are central to support the authors' claim that Pcgf6 works as a tumor suppressor. Also, to justify the search for a molecular mechanism. However, I find some inconsistencies in these data, which may reflect technical issues (although I am not sure). Confirming the *in vivo* phenotypes would thus be very important.

Finally, the already thorough genomic and molecular analyses could benefit from some further data analysis and additional experiments to understand how disruption of Pcgf6 alters (or not) the PRC1.6 complex in this model.

These are my detailed comments:

1- The survival curves for $E\mu\text{-Myc}$ mice (only) in Fig 1 and Fig S1 look quite different. For example, the median survival is 97 days in Fig S1A (in the experiment comparing survival in the Mga model), while it is 203 days in Fig 1 (Pcgf6 model). Instead, the median survival for the combinations with Mga versus Pcgf6 loss are quite similar, and perhaps statistically not significant. The median survival for the $E\mu\text{-Myc}$ model that can be inferred from other publications is also variable.

Resolving the discrepancy in the $E\mu\text{-Myc}$ reference model is critical to claim that Pcgf6 is working as a tumor suppressor.

This is an important issue that was brought up by both Reviewers: please see our detailed answer on this point in our reply to Reviewer #1 (point 1).

2- Minor: The analysis on potential differences in clonality as a measure of increased tumor initiation (Fig S2) is interesting but raises some questions. For example, the possibility that the percentage of purity in each tumor may be affecting the result. Was this taken into account? One could argue that these tumors are still oligoclonal.

Here, it is important to consider that we are scoring read counts for Immunoglobulin heavy chain mRNAs: by definition, these are expressed only in B-cells, making other cell types – albeit certainly present to various degrees – irrelevant in this analysis. As a corollary, the only relevant contaminant here would be non-tumoral B-cells. However, since classical DNA analyses in the same model led to the common wisdom that $E\mu\text{-myc}$ tumors are generally mono- or oligoclonal, we infer that contaminating B-cells must be – if any – a small minority in these tumor samples.

We have completed our text as follows regarding this point: “we exploited our RNA-seq profiles (see below) to analyze tumor clonality through the scoring of reads in the Immunoglobulin heavy chain variable region (Barbosa et al. 2020): in contrast with the widespread concept that lymphomas arising in Eμ-*myc* mice are monoclonal, classically based on Southern blotting (Adams et al. 1985) or PCR, we detected multiple clones in most samples (Fig. S2; Table S3), pointing to a more complex oligo- or polyclonal organization of these lymphomas. Most relevant here, our data did not reveal any consistent impact of the *Pcgf6* genotype on clonal complexity, indicating that accelerated tumor onset in the *Pcgf6*^{+/fl} and *Pcgf6*^{fl/fl} backgrounds could not simply be ascribed to increased clonality.”

3- Can the authors provide more information about the 78+65 differentially expressed genes in the tumor RNA-Seq analysis? This gene list doesn't seem to have been included in the Supplementary tables, and one wonders if using for example pathway analysis tools would yield any relevant findings.

As a note here, upon re-counting, the 78+65 differentially expressed genes are in fact 84+81. This changes nothing to our data and conclusions. We have updated those numbers this in the text and Fig. S3D.

We agree with the Reviewer on this point, and have added a dedicated paragraph to address this issue, with additional information in Table S4 and Figure S3. The insertion reads as follows:

“While few DEGs were called between *Pcgf6* KO and WT tumors (Fig. S3C), these genes might still be relevant to the more aggressive phenotype of the mutant tumors. Gene Ontology (GO) analysis of these DEGs (Fig. S3G) pointed out several functional categories, among which we noted several immune-related processes among the down-regulated genes. Whether these reflect true differences in gene expression in *Pcgf6* KO vs. WT tumor cells or differential infiltration by the host's immune system (e. g. antigen presenting cells, dendritic cells, macrophages, B- or T-lymphocytes) remains to be determined; nonetheless, these observations suggest that one possible mechanism by which *Pcgf6* suppresses lymphomagenesis may be through modulation of anti-tumoral immune responses.”

Also, restricting the differential analysis to those genes that are direct targets of *Pcgf6* (based on the ChIP-Seq data generated) could yield different results.

The main relevance of including our ChIP-seq data (Fig. S4B) is that they confirm two key features that were previously reported in other systems, and are thus confirmed in our lymphomas. Quoting from our text, in the section “*Mga*-dependent recruitment of *Pcgf6* to active chromatin” we first have the following introductory statement:

“At first sight, the limited impact of *Pcgf6* on transcriptional profiles appears at odds with its known function as a component of the PRC1.6 complex. The latter should depend on *Mga*, which is required both for the integrity of PRC1.6 and for the recruitment of *Pcgf6* to chromatin, as shown in mouse embryonic stem cells (mESCs) and lung tumor cells (Gao et al. 2012; Endoh et al. 2017; Stielow et al. 2018; Scelfo et al. 2019; Mathsyaraja et al. 2021).”

This is followed by the description of our data: “The distribution of ChIP-seq reads among annotated promoters and distal sites in the genome (Fig. S4B), confirmed two of key features established in previous studies. First, the *Pcgf6* signal observed in the control Eμ-*myc* lymphoma was lost not only in *Pcgf6*^{-/-}, but also in *Mga*^{-/-} cells, in line with the role of *Mga* in recruiting *Pcgf6* to chromatin. Second, *Pcgf6* did not co-localize with the PRC-associated marks H3K27me3 and H2AK119Ub, but showed preferential binding to active chromatin, as previously observed in mESCs (Stielow et al. 2018; Scelfo et al. 2019).”

The latter result is the most relevant here: indeed, our data show widespread *Mga*-dependent binding of *Pcgf6* (and thus presumably of the whole PRC1.6 complex) to active chromatin, in particular at promoters. This type of promiscuous cross-linking in ChIP-seq profiles is common with many transcriptional regulators (i. e. components of the basal transcriptional machinery, transcription factors or co-factors, etc...) and cannot be taken to indicate actual regulatory interactions. Combined with the lack of transcriptional alterations on pre-tumoral *Pcgf6* KO cells, we feel that trying to relate such widespread binding to the presence of few differentially expressed genes in tumor samples (which include both tumor and infiltrating cells) would be a questionable and potentially misleading exercise, from which no formal conclusion could be drawn.

4- It would be important to include in the discussion the fact that in DLBCL, *Mga* seems to score only as a potential tumor suppressor in ABC-like DLBCL, but not GCB-like. Emu-*Myc* models do not resemble DLBCL in major genetic or phenotypic traits, but if any, the resemblance to ABC-like DLBCL may be smaller than GCB.

This is a relevant point, which we have now treated in the text in a slightly wider context, as follows:

“Most importantly, our data do not formally rule out a role for *Mga*/*Max* and PRC1.6 in antagonizing *Myc*/*Max*-dependent transcription in other tumor types, including DLBCL. In particular, the combination of

Eμ-myc and *CD19-Cre*, targeting *Mga*^{fl/fl}, may not reproduce the more mature activated B-cell (ABC) DLBCL subtype in which *Mga* mutations were reported (Reddy et al. 2017) – although we note that a subset of *Eμ-myc* tumors do show ABC-like expression profiles (Schleich et al. 2020). Moreover, the oncogenic activation of *Myc*, as modeled by the *Eμ-myc* transgene, might conceivably be sufficient to overcome the repressive function of *Mga*: in ABC-type DLBCL, in which *MYC* translocation is relatively infrequent (Reddy et al. 2017; Bisso et al. 2019), *Myc* activity may well be antagonized by *Mga*, underlying the selective pressure to inactivate it. Resolving this question would imply the joint scoring of *MYC* translocations and *Mga* mutations in a sizeable number of DLBCL samples, well above those reported so far (125 cases, of which 42 with *MYC* rearrangements and 7 with *MGA* mutations) (Reddy et al. 2017). Finally, any of the five *Mxd/Mnt* proteins that form alternative dimers with *Max* may also contribute repressive activity on common *Mga*- and *Myc*-target genes, and the balance between all these factors may differ between cell/tumor subtypes, experimental models and/or clinical cases.

Altogether, the contribution of the *Mga/Max-PRC1.6* complex to DLBCL pathogenesis remains to be addressed. This notwithstanding, our data in the *Eμ-myc* model establish that in conditions in which *Mga* shows no obvious impact, *Pcgf6* deletion clearly accelerates *Myc*-induced lymphomagenesis”

As noted in the aforementioned text, the numbers of DLBCL cases in which *MYC* translocations and *MGA* mutations have both been scored does not allow a formal conclusion. In particular, Reddy et al. (2017) addressed *MYC* translocations by FISH on a subset of 125 samples. Considering those samples with and without *MYC* translocations, *MGA* mutations were present in 5/42 and 2/83 cases, respectively. This difference remains below statistical significance; as such, we feel that it would be misleading to discuss this in more detail in our paper.

Finally, we shall note here that Schleich et al. (2020) identified lymphomas with either ABC- or GCB-like features in the *Eμ-myc* model. Nonetheless, the relevance and pathological implications of the GCB/ABC profiles in those mouse lymphomas relative to human DLBCL remains to be unraveled. A more detailed discussion of this issue would be beyond the scope of our work.

5- The effects of *Pcgf6* loss on *Myc*-dependent apoptosis in *Emu-Myc* B cells (Fig. 2) do not seem associated with haploinsufficiency, this is, there is no loss of apoptosis in *Pcgf6* del/+ B cells despite these animals have increased tumor incidence and shortened latency. That would mean that this is perhaps not a major mechanism by which *Pcgf6* loss contributes to tumorigenesis (which is opposite to the author's argument).

The Reviewer is right here: while we record significant effects of homozygous *Pcgf6* loss on B-cell counts (Fig. 2A) and apoptosis (Fig. 2B), this is not observed for the heterozygous mutant. This is why we had referred explicitly to the homozygous in *Eμ-myc; CD19-Cre; Pcgf6*^{fl/fl} mice in our conclusions and introduced a note of caution with “at least in part”. We have now added this additional statement:

“Of note, while the effect of *Pcgf6* loss on *Myc*-induced lymphomagenesis was already apparent in heterozygous *Pcgf6*^{fl/fl} mutant mice (Fig. 1A), the same was not true for B-cell counts and apoptosis (Fig. 2A, B): this apparent discrepancy might be due either to a limiting sensitivity of the pre-tumoral measurements, or to the co-existence of additional mechanisms of tumor suppression by *Pcgf6*.”

6- In the same Fig 2, the authors show data on cell cycle analysis. There do not seem to be major differences, but one wonders if the fraction of cells in cycle (Ki67+) is affected by *Pcgf6* loss (since many B cells in the bone marrow are resting).

The reviewer is correct in pointing out the quiescent state of most B cells in the bone marrow of control non-transgenic mice. However, our data clearly show that the *Eμ-myc* transgene induces proliferation in this compartment, with a significant increase in S-phase cells, but with no significant effect of *Pcgf6* loss (whether homo- or heterozygous: Fig. 2C). In the data shown here, we have used a pulse of EdU incorporation *in vivo* and 2D-FACS analysis to quantify S-phase cells, which provides a more direct and quantitative readout than Ki67 in measuring cycling cells. On this basis, we feel that these data provide adequate support to our conclusion that “loss of *Pcgf6* caused no major alterations in the cell cycle profiles of B220⁺CD19⁺ B-cells”, without the need to produce additional data on Ki67.

It would also be important to show representative cell cycle plots to help understand why the low percentage of cells in G2/M.

As requested by the Reviewer, we have now added representative FACS profiles in Fig. 2C. The data imply that these cells have an extremely short G2, and must in fact be entering mitosis as soon as completing S-phase. In further support of this interpretation, note that a 2h pulse of EdU *in vivo* was sufficient for part of the cells to incorporate this nucleotide and complete mitosis, and thus be back in G1 (i. e. 2N DNA content) while scoring as EdU-positive. We have added a note in this regard to the legend of Fig. 2C.

Finally, we note here that previous studies also scored low numbers of G2/M cells in the same model [e. g. Rempel et al. 2009 PLOS Genetics 5(9): e1000640; D'Artista et al. 2016 Oncotarget 7, 21786-98;

Vandenberg et al. 2016 *Cell Death Dis.* 7, e2046; Vecchio et al. 2019 *Cell Death Dis* 10, 320]. This feature is thus well-established. However, as this is not directly pertinent to the question addressed in this experiment, we do not deem this relevant of to discuss it further in our text.

7- An important part of this study is the comparison between the consequences of Mga and Pcgf6 loss. These two factors are components of the PRC1.6 complex. Loss of Mga disrupts the complex (previous studies and this one) and prevents Pcgf6 binding to chromatin. It is unclear, however, what happens to Mga when Pcgf6 is lost. Zhao et al (J Biol Chem 2017, 292) reported that in ES cells, loss of Pcgf6 results in a partial disruption of the PRC1.6 complex, although Mga, Max and L3mbtl2 remain bound to chromatin in a smaller complex. There were also some specific changes in gene expression but no effects on H2AK119Ub levels. Based on these observations and the current data, it would be important to investigate the composition of the PRC1.6 complex in the Emu-Myc system upon Pcgf6 is lost (e.g., co-IP).

Complementary to the above, it would be important to determine the distribution of Mga on chromatin (at least via immunoblot of chromatin-enriched fractions, if ChIP-Seq is not feasible). This would be a helpful molecular insight to understand the different outcomes of Mga and Pcgf6 loss in vivo.

We now cite Zhao et al. in our Introduction, highlighting the point brought up by the reviewer, in particular regarding the role of Pcgf6 in the full assembly of the PRC1.6 complex on chromatin, but not for the binding of Mga/Max and L3mbtl2. Our text now reads as follows: "...depletion of Pcgf6 caused dissociation of several subunits (Ring1A/B and Rybp) while others (Mga, Max and L3mbtl2) remained chromatin-bound (Zhao et al. 2017). Altogether, these findings suggest that Mga/Max and Pcgf6 contribute to the hierarchical assembly of the PRC1.6 complex onto chromatin, may thereby counteract transcriptional activation by Myc and E2F at common target genes, and thus also their growth-promoting and oncogenic activities."

This being said, while all of the issues raised here by the reviewer are pertinent, and were all considered in the course of our work, the reagents that we were able to test for Mga did not allow us to address any of these questions in a systematic and conclusive manner (neither by ChIP, nor co-IP, nor immunoblotting). On this basis, after substantial effort, **we decided to draw the line and submit the available data, as we believe that these constitute a coherent package.** Most importantly in this regard, **the main conclusions made in our manuscript are all within the limits of what can be formally supported based on the available data, and do not depend on the additional issues raised here.**

In particular:

- (i.) As outlined by the Reviewer, **loss of Mga disrupts the PRC1.6 complex and prevents Pcgf6 binding to chromatin.** Indeed, as explained in more detail above (point 3), our ChIP-seq data confirm this well-established fact. In other words, as previously shown in other systems, Mga is required for the recruitment of Pcgf6 to its target loci in Eμ-*myc* lymphomas.
- (ii.) The reviewer insists on the need address **what happens to Mga**, as well as to **the composition of the PRC1.6 complex when Pcgf6 is lost.** While we agree with the general interest of these questions, we do not have the tools to address them; furthermore, their actual relevance has become rather secondary in the actual context of our current manuscript, since Mga – unlike Pcgf6 – did not score as a tumor suppressor in our experiments.

Thus, our data are in line with the well-established role of Mga in the PRC1.6 complex (as assessed by Pcgf6 ChIP-seq). Yet, unlike Pcgf6, deletion of Mga has no obvious impact on Myc-induced lymphomagenesis, reinforcing the notion that we are looking at a different activity of Pcgf6.

Altogether, and with all due caution, we believe that the conclusions made in our text on the distinct tumor suppressor activity of Pcgf6 are duly supported by our data. On the other hand, pursuing the activities of MGA and PRC1.6 in DLBCL is an important goal, but beyond the scope of our paper.

8- ChIP-Seq data (Fig S4): the data shows that loss of Pcgf6 results in a (significant) increase in H3K27Ac reads. The authors do not discuss this in the manuscript, but this effect seems quite prominent, extends to both promoters and distal sites, and may have some conceptual implications. Any insights?

The reviewer is correct in pointing out the generally elevated **H3K37me3-H3K27Ac** signal in the Pcgf6-null sample. For technical reasons, we believe that this is a result that remains preliminary, and should thus be interpreted with all due caution. We have completed our text as follows:

"Of note here, one of the apparent changes observed in the *Pcgf6*^{-/-} lymphoma was an increase in the **H3K27Ac-H3K37me3** signal on chromatin, at both proximal and distal sites (Fig. S4B). However, owing to the small number of Pcgf6- and Mga-null lymphoma cell lines available in our work, as well as to the limiting availability of compound Eμ-*myc*; *Pcgf6*^{fl/fl} mice (Table S1), which precluded ChIP-seq analysis in pre-tumoral B-cells (Sabò et al. 2014), we could not determine whether this reflected a real effect of Pcgf6 on **H3K27Ac-H3K37me3**, or a spurious difference - possibly due to clonal variability among lymphomas. For the

same reasons, we were unable to address whether loss of PRC1.6 activity might impact Myc's binding profiles in our model. This scenario appears unlikely, however, given that Pcgf6 loss showed no significant impact on Myc-associated gene expression profiles (Fig. 3, Fig. S3)."

Most importantly, none of this impacts on the two key features highlighted by our ChIP-seq profiles (see above, point 3).

March 29, 2022

RE: Life Science Alliance Manuscript #LSA-2021-01344-TR

Dr. Bruno Amati
European Institute of Oncology (IEO)
Dept. of Molecular Oncology and Center for Genomic Science
Via Adamello, 16
Milan, Milan 20139
Italy

Dear Dr. Amati,

Thank you for submitting your revised manuscript entitled "Polycomb Group Ring Finger Protein 6 suppresses Myc-induced lymphomagenesis". We would be happy to publish your paper in Life Science Alliance pending final revisions necessary to meet our formatting guidelines.

- Please address Reviewer 2's final comments in the last paragraph
- supplementary references should be part of the main manuscript references
- supplementary materials and methods should be part of the main manuscript Materials & Methods section
- Please upload all figure files as individual ones, including the supplementary figure files; all figure legends should only appear in the main manuscript file
- please add the Twitter handle of your host institute/organization as well as your own or/and one of the authors in our system
- please make sure the author order in your manuscript and our system match
- please consult our manuscript preparation guidelines <https://www.life-science-alliance.org/manuscript-prep> and make sure your manuscript sections are in the correct order
- please add molecular weights next to all blots

A. FINAL FILES:

B. MANUSCRIPT ORGANIZATION AND FORMATTING:

Sincerely,

Reviewer #1 (Comments to the Authors (Required)):

This revised manuscript has addressed most concerns of the reviewers through carefully modifying the text to better match the data, providing additional experimental characterizations such as FACS in Figure 2C, and citing pertinent studies. The authors also emphasized the goals and technical limitations of the study, which although it could not fully define the role of MGA in DLBCL, did open interesting perspectives on the multifaced MGA, PCGF6, and PRC1.6 functions in different systems, warranting future studies on similar topics. Specifically, this study underscores the need for additional animal models for studying the initiation and progression of DLBCL, which is certainly a challenge given the vastly heterogeneous genetic landscapes present in clinic. Overall, I think that this paper will stimulate further research and I support publication of this revised version.

On a minor note, if the authors were to further pursue the PRC1.6 interaction with the chromatin without concern about the promiscuous crosslinking by X-ChIP, they should consider the published "Cut-and-Run" approach.

Reviewer #2 (Comments to the Authors (Required)):

I want to thank the authors for the detailed referenced answers to the comments raised during the review of the manuscript, as well as the additional figure panels and the several edits in the main text. The changes in this revised version of the manuscript help clarify most of the questions raised. The extensive discussion on the similarities and differences between the Emu-Myc model and human DLBCL is also greatly appreciated.

The results presented in this study will likely be a primer for future investigation. Although the thorough exploration of potential mechanisms resulted in most cases in negative results, evidence for a tumor suppressor role of Pcgf6 independent of transcriptional or apparent epigenetic alterations seems clear.

Some observations are particularly intriguing and grant future investigation. Particularly, one wonders how loss of Pcgf6 chromatin binding associates to different phenotypes (i.e. tumor suppressor effect) depending on whether such loss is a consequence of Mga depletion vs direct Pcgf6 manipulation. The only difference detectable in these two scenarios is a widespread change in H3K27Ac, obvious for Pcgf6 deletion but absent in Mga -/- samples, which should be taken with caution because of the limited number of samples analyzed, as well noted by the authors. One intriguing possibility is that loss of Mga

may have some compensatory effect on the effects of Pcgf6 loss (unclear how that would work). I guess that future research would be able to explain this paradox, particularly upon access to better tools to study Mga and the through the analysis of larger datasets.

Finally, just to point out some confusion regarding point #8 discussed in my previous review:

In my comments I referred to the changes in H3K27Ac signal shown in the ChIP-Seq data presented in Fig S4B. In their response, the authors now mention H3K37me3 instead, not H3K27Ac. This is confusing, as it does not seem to fit with the data presented or the antibodies listed for the experiment and appears to be a typo/mistake -- unless there is something that I am missing. If a typo, it should be corrected. Otherwise, some additional clarification would be very helpful.

-

April 4, 2022

RE: Life Science Alliance Manuscript #LSA-2021-01344-TRR

Dr. Bruno Amati
European Institute of Oncology (IEO)
Dept. of Molecular Oncology and Center for Genomic Science
Via Adamello, 16
Milan, Milan 20139
Italy

Dear Dr. Amati,

Thank you for submitting your Research Article entitled "Polycomb Group Ring Finger Protein 6 suppresses Myc-induced lymphomagenesis". It is a pleasure to let you know that your manuscript is now accepted for publication in Life Science Alliance. Congratulations on this interesting work.

DISTRIBUTION OF MATERIALS:

Again, congratulations on a very nice paper. I hope you found the review process to be constructive and are pleased with how the manuscript was handled editorially. We look forward to future exciting submissions from your lab.

Sincerely,
